# Preparation and Properties of a High-Performance EOEOEA-Based Gel-Polymer-Electrolyte Lithium Battery

**DOI:** 10.3390/polym11081296

**Published:** 2019-08-02

**Authors:** Wenwen Ding, Chun Wei, Shiqi Wang, Linmin Zou, Yongyang Gong, Yuanli Liu, Limin Zang

**Affiliations:** 1College of Materials Science and Engineering, Guilin University of Technology, Guilin 541004, China; 2Key Laboratory of New Processing Technology for Nonferrous Metals and Materials, Ministry of Education, Guilin 541004, China

**Keywords:** lithium battery, GPE, EOEOEA, deep in-situ polymerization

## Abstract

Gel polymer electrolyte (GPE) is a promising candidate for lithium-ion batteries due to its adhesion property (like a solid), diffusion property (like a liquid), and inhibition of the growth of lithium dendrite. In this paper, 2-(2-ethoxyethoxy)ethyl acrylate (EOEOEA) and LiBF_4_ electrolyte were mixed as precursors of gel polymer electrolytes. Through thermal curing, a thermally stable GPE with high ionic conductivity (5.60 × 10^−4^ s/cm at 30 °C) and wide room temperature electrochemical window (4.65 V) was prepared, and the properties of the GPE were measured by linear sweep voltammetry (LSV), AC impedance spectroscopy, Thermogravimetric analysis (TG), and X-ray diffraction (XRD) techniques. On the basis of the in-situ deep polymerization on a LiFePO_4_ electrode and cellulose membrane in a battery case, EOEOEA-based GPE could be derived on both LiFePO_4_ electrode and cellulose membrane. Meanwhile, the contact between GPE, LiFePO_4_ electrode, and lithium electrode was promoted. The capacity retention rate of the as-prepared LiBF_4_-EOEOEA 30% gel lithium battery reached 100% under the condition of 0.1 °C after 50 cycles, and the Coulombic efficiency was over 99%. Meanwhile, the growth of lithium dendrite could be effectively inhibited. GPE can be applied in high-performance lithium batteries.

## 1. Introduction

Due to the popularization of portable mobile electronic devices and electric vehicles [1,2], high-performance lithium batteries and their safety have drawn increasing attention. The organic solvent in traditional liquid lithium batteries evaporates and leaks. The lithium dendrite growing in charge-discharge cycles can puncture the membrane, causing short circuits in batteries. At the same time, the heat released during the cycles can cause safety threats such as thermal runaway, or even spontaneous combustion [3]. In view of these issues, polymer lithium batteries, using polymer electrolytes instead of traditional organic liquid electrolytes, are free of leakage of organic liquid and growth of lithium dendrite owing to the isolation function of polymer electrolytes, making polymer lithium batteries promising candidates in the future [4,5].

According to the presence or absence of plasticizer, polymer electrolytes are divided into solid polymer electrolytes (SPEs) and gel polymer electrolytes (GPEs), respectively. Polymer electrolytes should possess the following properties: (1) High room-temperature lithium-ion conductivity to ensure fast charge-discharge processes; (2) high chemical, electrochemical, and thermal stability—the premise for the safety and stability of the batteries; and (3) strong mechanical strength to support the assembly of lithium batteries [6].

At present, the ionic conductivity of SPEs is too low to meet the requirements of lithium batteries [7]. Moreover, an SPE contacts with cathodes and Li electrodes in a solid-solid mode, where the absence of wettability on the contact surface readily leads to voids, which greatly increase the internal resistance of lithium batteries and further leads to the deterioration of cycling performances of batteries [8]. Huang S et al. applied in-situ polymerization to effectively alleviate this problem in SPE lithium batteries, and 91% of the capacity was retained after 100 cycles [9]. Tong Y et al. prepared an interpenetrating network polymer electrolyte with ionic conductivity more than 1 × 10^−5^ s/cm at room temperature and an electrochemical window wider than 4.5 V. The assembled all-solid-state lithium battery possessed a Coulomb efficiency of 99% after 200 cycles, showing excellent performance [10]. Zeng X et al. also prepared a polymer electrolyte with an interpenetrating network structure, which greatly inhibited the crystallization of Polyethylene oxide (PEO), and the ionic conductivity was improved; after 200 cycles, 85% of the capacity of the all-solid-state lithium battery was retained [11].

GPE is a special material, which has the properties of both adhesion (like a solid) and diffusion (like a liquid). Lithium dendrite is difficult to grow in GPE lithium batteries, therefore the safety and cycling life of lithium batteries can be greatly improved. Currently, GPE has been extensively studied [12]. Dong X et al. obtained refractory GPE materials by using Li_6.4_Ga_0.2_La_3_Zr_2_O_12_ as ion-conductive in GPE [7]. Jimin S et al. combined ionic liquid and GPE for the first time, and the obtained GPE showed resistance to fire. The battery had a specific discharge capacity of about 120 mAh/g after 300 cycles, showing excellent recycling performance and safety [13,14]. Karuppasamy. K et al. proposed the concept of a novel ternary gel polymer electrolyte. After 100 cycles, the discharge specific capacity was about 120 mAh/g, and 80% of the capacity was retained, showing better charge/discharge performance [15].

Modern polymer lithium batteries are produced with a simple casting method [16,17]. Typically, an electrolyte precursor is poured into a Polytetrafluoroethylene or glass mold in a glove box, and the excess organic solvent is evaporated at a high temperature to form a thin film for the assembly of a lithium battery. However, because of the surface tension between the precursor electrolyte and Polytetrafluoroethylene mold, the resulting membrane is subject to air bubbles, shrinkage, and uncontrollable thickness. When a battery is assembled, voids are easy to generate between the film and solid electrodes, greatly increasing the internal resistance and shortening the cycling lives of the batteries. We cannot ignore the impact of volatile organic solvents on the glove box environment at the same time.

Considering the poor contact between polymer electrolytes and solid electrodes, Chen X and co-workers [18] proposed an in-situ polymerization technique on LiFePO_4_ electrodes. In detail, a polymer precursor electrolyte was infiltrated into LiFePO_4_ electrodes in air. The mixture was heated to initiate polymerization. Then, the as-treated electrodes were sliced for assembly. In this technique, the polymer electrolyte is able to fully contact the active material in the electrodes, but the electrode slices have burrs on the edge, still leading to poor contact and increase of resistance. Meanwhile, the lithium salts in the precursor electrolyte can absorb water in air, probably affecting the occurrence of polymerization. Cui Y proposed another in-situ polymerization technique on LiFePO_4_ electrodes in a battery container. However, because only one layer of polymer electrolyte was left on the surface of the electrode, the active materials could not fully contact the polymer electrolyte, resulting in incomplete charging and discharging of the lithium battery [19].

Guo et al. directly coated a polymer electrolyte on the surface of lithium electrodes via UV-curing. This method is limited by the surface tension between the precursor electrolyte and the Li electrode. If the Li electrode is not completely covered by the polymer electrolyte membrane, short circuits take place [11].

Compared with the method of immersing polypropylene film (PP) in an electrolyte to prepare a composite electrolyte membrane [20], we propose to inject the precursor electrolyte into the LiFePO_4_ electrode and the cellulose membrane for deep in-situ polymerization, to derive GPE simultaneously on LiFePO_4_ electrodes and cellulose membranes where the contact between them could be greatly promoted. This method is simple and efficient. Furthermore, the physical, chemical, and electrochemical properties of EOEOEA-based GPE and relative EOEOEA-based GPE lithium batteries are studied for the application of novel polymer lithium batteries.

## 2. Materials and Methods

### 2.1. Materials

Main experimental materials: LiFePO_4_ (Kejing Material Technology Co., Ltd., Hefei, China), N-methyl pyrrolidone (NMP, Xilong Chemical Co., Ltd., Shantou, China), conductive carbon black (Kejing Co., Ltd., Shenzhen, China), EOEOEOEA (Aladdin Biochemical Technology Co., Ltd., Shanghai, China; average molecular weight = 600, dried with molecular sieves), benzoyl peroxide (BPO, Aladdin Biochemical Technology Co., Ltd., Shanghai, China), lithium bisoxalate (LiBOB), lithium difluorooxalate borate (LiDFOB), lithium tetrafluoroborate (LiBF_4_), lithium bistrifluoromethyl sulfimide (LiTFSI), lithium hexafluorophosphate (LiPF_6_), ethylene carbonate (EC), vinylene carbonate (VC), dimethyl carbonate (DMC), diethyl carbonate (DEC, Dadao New Material Technology Co., Ltd., Huizhou, China), Polythylene terephthalate film (90 μm thick, Lvcheng Electrical Machinery Co., Ltd., Shanghai, China), and cellulose membranes (NKK Company, Kochi, Japan).

### 2.2. Preparation of LiFePO_4_ Electrode

To start, 13.18 g of NMP solution was placed in a 20 mL beaker. Then, 9.2 g of LiFePO_4_ powder, 0.4 g of PVDF, and 0.4 g of conductive carbon black were gradually and successively added to the beaker, along with stirring with a high-speed mixer. The slurry was stirred for 2 h for uniformity. During the stirring, the beaker was sealed with a film to prevent PVDF from deliquescence. Afterwards, the uniform slurry was coated on tin foil to prepare a 100 μm-thick electrode. For drying, the LiFePO_4_ electrode was first placed in an oven at 60 °C for 20 min to dry the surface, and was then placed in a vacuum oven at 90 °C for a night to dry completely. The dried LiFePO_4_ electrode was cut into 16 mm-diameter wafers. The mass of each wafer was about 0.0175–0.0185 g, and that of the active species was about 0.0161–0.0170 g. The LiFePO_4_ wafers absorb water when exposed to air, so they need to be dried at 90 °C for 3 h before being placed in the glove box. 

### 2.3. Preparation of EOEOEA-Based GPE

In this experiment, EOEOEA-based GPE was prepared with a thermal curing method. All the materials required were stored in the glove box and were protected with argon gas. The entire experiment was carried out in the glove box as follows:

#### 2.3.1. Preparation of 1 mol/L LiBF_4_ Electrolyte

LiBF_4_ was dissolved in a mixture solution (containing 2 vol% VC) of EC/DEC/DMC at a volumetric ratio of 1:1:1. The mixture was thus a traditional 1 mol/L LiBF_4_ electrolyte.

Similarly, 1 mol/L LiBOB, LiDFOB, LiTFSI, and LiPF_6_ electrolytes were prepared.

#### 2.3.2. Screening of Lithium Salts

The anions of lithium salts influence the chemical, electrochemical, and physical properties of GPE, and ultimately affect the performance of the GPE lithium battery [21,22]. In this paper, EOEOEA was selected as the gel polymer substrate. LiBOB, LiDFOB, LiTFSI, LiBF_4_, and LiPF_6_ were selected as the lithium salts to prepare different lithium-salt‒EOEOEA GPEs. In detail, 0.60 g of LiBOB electrolyte was mixed with 0.30 g of an EOEOEA solution (50% of the LiBOB electrolyte in mass) and 0.0045 g of BPO (0.5% of the LiBOB electrolyte and EOEOEA in mass). Polymerization was initiated after heating and LiBOB-EOEOEA 50% GPE was obtained. Similarly, LiDFOB-EOEOEA 50% GPE, LiTFSI-EOEOEA 50% GPE, LiBF4-EOEOEA 50% GPE, and LiPF6-EOEOEA 50% GPE were prepared. Based on the ionic conductivity and electrochemical window of the five EOEOEA GPEs, the most suitable lithium salt for EOEOEA was LiBF_4_.

#### 2.3.3. Preparation of LiBF_4_-EOEOEA GPEs

To start, 0.6 g of the LiBF_4_ electrolyte was mixed with 0.18 g of the EOEOEA solution (30% of the mass of LiBF_4_ electrolyte) and 0.004 g of BPO. The mixture was the precursor electrolyte, which was then heated at 75 °C for 12 h to derive a gel polymer electrolyte, denoted by LiBF_4_-EOEOEA 30%. Similarly, GPEs of LiBF_4_-EOEOEA 40%, 50%, and 60% were prepared.

### 2.4. Characterizations of GPE

Fourier infrared spectroscopy (FTIR) analysis of the gel polymer electrolytes was performed with an NICOLETNEXUS470 spectrometer (Perkin-Elmer, The States, Waltham, MA, USA). The test wavelength range was 4000–400 cm^−1^.

Field emission scanning electron microscopy (FESEM) analysis of the original LiFePO_4_ electrodes, surfaces and cross-sections of LiFePO_4_ electrodes after battery cycling, original cellulose membrane, cellulose composite membranes after battery cycling, original lithium electrodes, and lithium electrodes after battery cycling was performed with an S-4800 field emission scanning electron microscope (HITACHI Company, Tokyo, Japan). The acceleration voltage was 5 kV and samples were coated with gold for 30 s.

Thermogravimetric analysis (TG) of gel polymer electrolytes was performed with a Q-500 integrated thermal analyzer (TA Company, The States, Newcastle, DE USA). The samples were first dried overnight in a vacuum oven at 80 °C. Then, the samples were heated in a N_2_ atmosphere furnace with a temperature range of 20–800 °C at a heating rate of 10 °C/min.

X-ray diffraction (XRD): The powder of original LiFePO_4_ electrodes and the powder of LiFePO_4_ electrodes after battery cycling were performed with an X’Pert PRO X-ray diffractometer (PANalytical, Almelo, The Netherlands). 2θ of 5°–80° was scanned.

### 2.5. Characterizations of Electrochemical Properties of GPE

#### 2.5.1. Electrochemical Stability

The voltage that the electrolyte can withstand is limited, otherwise the electrolyte will decompose due to chemical reactions. The electrochemical window represents the voltage range in which the electrolyte can be stably present. Electrochemical window is an important index to the electrochemical stability of an electrolyte, and can be measured by linear sweep voltammetry (LSV).

In the present work, a 90 μm-thick PET film was shredded, with a microtome, into a ring film with an inner diameter of 10 mm and an outer diameter of 16 mm. The inner circle is the contact area between the GPE and electrodes. The ring was sonicated for 30 min, and was then dried in a vacuum oven at 70 °C overnight. After drying, the ring was fast transferred to a glove box for reservation.

In the glove box filled with argon gas, a 1 mm-thick stainless steel sheet was placed into a container on the anode side. The PET ring film was placed on a stainless steel sheet, and then a precursor electrolyte was dropped into the inner circle of the PET ring until the hole was filled up. Then, a new lithium sheet and anode container were inserted. All these components were sealed at 6 MPa to compose a stainless steel sheet/GPE/Li button battery, as shown in Figure 1. Then, the quasi-battery was heated at 75° C for 12 h until the precursor electrolyte was polymerized. The stainless steel functioned as the working electrode, and the lithium sheet served as the reference and counter electrode.

The as-treated battery was tested by LSV with a CHI660 electrochemical workstation, Chenhua Company, Shanghai, China The scanning rate was set to be 1 mV/s and the scanning voltage was in the range of 0.5–6.5 V.

#### 2.5.2. Conductivity

Similarly, following the method stated in Section 2.5.1, a stainless steel sheet/GPE/stainless steel sheet button battery was assembled, and the AC impedance measurements of this battery at different temperatures was performed with the electrochemical workstation. The test frequency was in the range of 100 mHz–1 MHz and the amplitude was 5 mV. Zview software was used for the fitting of the AC impedance spectra to derive the values of bulk resistance, which were used to calculate conductivity according to Equation (1).
(1)$$\sigma =\frac{L}{R*A}$$
where the thickness of GPE is L in units of cm; the bulk resistance of GPE is R in units of Ω; and A is contact area, which is the area of the inner circle of the PET membrane in this experiment, in units of cm^2^. Thus, the conductivity σ at different temperatures would be calculated. Accordingly, the activation energy required for the movement of lithium ions could be also calculated according to the Arrhenius equation, Equation (2).
(2)$$\sigma =\mathrm{Aexp}\left(-\frac{\mathrm{Ea}}{\mathrm{RT}}\right)$$
where A represents the pre-exponential factor; Ea stands for activation energy, kJ/mol; R is the ideal gas constant 8.314 J·mol^−1^·K^−1^; and T is the temperature (K) in the test.

#### 2.5.3. Assembly of GPE Lithium Batteries

Using the LiFePO_4_ sheet as the cathode, Li sheet as the anode, and the GPE cellulose composite membrane as the electrolyte and isolation membrane, we assembled EOEOEA-based GPE batteries. In detail, a LiFePO_4_ sheet was placed in the middle of a container on the cathode side. Three drops of EOEOEA 30% precursor electrolyte were added to the LiFePO_4_ sheet and then it was left to maintain for 2 h to make the precursor electrolyte fully penetrate into the LiFePO_4_ sheet. An original cellulose isolation membrane was inserted and was fully infiltrated with precursor electrolyte. Then, a new lithium sheet and nickel foam were inserted. The anode container was assembled. The quasi battery was sealed at 5.5 MPa with a button battery sealer. Considering the fact that the polymerization of the precursor electrolyte on the cellulose membrane is favorable to effective control of the thickness of the gel polymer electrolyte membrane [23], the as-assembled quasi battery was heated at 75 °C for polymerization. The as-prepared battery was denoted as an EOEOEA 30% lithium battery. Similarly, LiBF_4_-EOEOEA 40%, LiBF_4_-EOEOEA 50%, and LiBF_4_-EOEOEA 60% lithium batteries were assembled.

#### 2.5.4. Assembly of Traditional LiBF_4_ Electrolyte Liquid Lithium Batteries

Using the LiFePO_4_ sheet as the cathode, Li sheet as the anode, and cellulose membrane as the battery membrane, LiBF_4_ electrolyte liquid lithium batteries were assembled. In detail, a LiFePO_4_ sheet was placed in the middle of a container on the cathode side and then was covered with a cellulose membrane. Three drops of LiBF_4_ electrolyte were added onto the cellulose separator. Then, a lithium sheet and a nickel foam were inserted. The anode container was assembled. The quasi battery was sealed at 5.5 MPa with a button battery sealer. The assembled liquid lithium battery was removed from the glove box and was maintained overnight, so that the LiBF_4_ electrolyte could fully penetrate the LiFePO_4_ electrode and cellulose membrane.

#### 2.5.5. Electrochemical Performances of Lithium Batteries

The charge-discharge cycling stability is a common index in the performance of a battery. In this study, a Neware high-performance battery detection system was used to measure the cycling and rate performances of those EOEOEA-based lithium batteries. The AC impedance of those batteries before and after the charge-discharge cycling was measured to study the interface stability of those batteries.

## 3. Results and Discussions

### 3.1. Screening of Lithium Salts

Figure 2a indicates the ionic conductivity of different lithium salt-EOEOEA 50% GPEs at 30 °C. Among these five GPEs, the LiBF_4_-EOEOEA 50% GPE showed the highest ion conductivity, about 1.23 × 10^−4^ S/cm, resulting from the combination of its excellent ionic mobility and dissociation constant and the fact that LiBF_4_ has the lowest charge transfer resistance among electrolytes [24,25]. LiTFSI exhibited an ionic conductivity of about 1.17 × 10^−4^ S/cm because LiTFSI could reduce the crystal region in the EOEOEA polymer. At the same time, LiTFSI-GPEs could corrode aluminum foil, thus limiting its applications. LiPF_6_ yielded non-conductive LiF during heating, so LiPF_6_-EOEOEA 50% GPE showed the lowest ion conductivity. It is well known that LiDFOB generates LiPF_6_ and LiBF_4_ during heating, so the ionic conductivity of LiDFOB-EOEOEA 50% GPE was between those of LiPF_6_ and LiBF_4_ [19]. 

Figure 2b shows the room-temperature electrochemical windows of different lithium salt-EOEOEA 50%-based GPEs. Among them, LiBOB-EOEOEA 50% GPE and LiPF6-EOEOE 50% GPE exhibited relatively narrow windows. LiBF4-EOEOEA 50% GPE possessed the widest electrochemical window of about 5.05 V, much wider than the values of traditional liquid lithium batteries and PEO-based solid polymer electrolytes (~3.9 V). Moreover, the electrochemical window profiles of LiDFOB-EOEOEA 50%, LiTFSI-EOEOEA 50%, LiBF4-EOEOEA 50%, and LiPF6-EOEOEA 50% GPE exhibited a spike at around 0.25 V, indicating the occurrence of oxidation reactions, in which lithium ions immigrated from lithium sheets to stainless steel sheets [19]. However, the high hardness of LiBOB-EOEOEA 50% GPE made it difficult for lithium ions to migrate. Therefore, no spike was observed at around 0.25 V for LiBOB-EOEOEA 50% GPE, as shown in Figure 2b. Considering the high ionic conductivity and wide electrochemical window, LiBF4 was selected as the lithium salt for batteries in the following sections. 

### 3.2. Mechanism of Polymerization of EOEOEA and LiBF_4_ Electrolyte

Figure 3a shows the polymerization of EOEOEA initiated by BPO. Figure 3b shows the precursor electrolytes with different contents of EOEOEA. Figure 3c shows the gel polymer electrolytes with different contents of EOEOEA. As shown in Figure 3a, under the heating condition, BPO decomposed to generate free radicals, causing polymerization of C=C in EOEOEA, and finally giving a GPE [26]. C=O had a higher affinity for BF_4_^−^, resulting in the large conductivity of LIBF_4_-EOEOEA GPE. Figure 3c shows that LiBF_4_-EOEOEA 10% and LiBF_4_-EOEOEA 20% precursor electrolytes were still liquids after the heating. This is because the content of the EOEOEA monomer was too small, and the EOEOEA gel polymer electrolytes that formed could not contain all the organic electrolytes. However, along with the increase of EOEOEA content, the degree of gelation increased. Transparent GPEs were successfully obtained after the heating of LiBF_4_-EOEOEA 30%, LiBF_4_-EOEOEA 40%, LiBF_4_-EOEOEA 50%, and LiBF_4_-EOEOEA 60% precursor electrolytes, and the hardness of the GPEs obtained were gradually increased in the same order.

Figure 4a shows the FTIR spectra of LiBF_4_-EOEOEA 30% precursor electrolyte before heating and LiBF_4_-EOEOEA 30% GPE after heating, and Figure 4b is an enlarged view of 2000–500 cm^−1^ in Figure 4a. In the spectrum of the precursor electrolyte, the band at 1730 cm^−1^ corresponds to the coupling of vibration of two carbonyl groups in BPO anhydride. The bands at 780 and 720 cm^−1^ correspond to the deformation vibration of C=C-H groups in EOEOEA monomer [27]. After heating and polymerization, LiBF_4_-EOEOEA 30% GPE was yielded. In the spectrum of the LiBF_4_-EOEOEA 30% GPE, the bands at 780 and 720 cm^−1^ (C=C-H) and band 1730 cm^−1^ completely disappeared, proving that the BPO-initiated polymerization of EOEOEA30% precursor electrolyte could take place. The bands at 2971, 1806, 1625, 1382, 1110, and 854 cm^−1^ correspond to the stretching vibration of CH_2_, symmetric stretching of C=O and asymmetric stretching of C=O, asymmetric stretching of C-O-C, vibration of C=O-O vibration and bending vibration of B–F, respectively. All these bands were observed in the precursor electrolyte and LiBF_4_-EOEOEA 30% GPE. 

### 3.3. Electrochemical Properties of LiBF_4_-EOEOEA-Based GPEs

The conductivities of GPEs with different EOEOEA contents at different temperatures are shown in Figure 5a. The conductivity of EOEOEA 30%, EOEOEA 40%, EOEOEA 50%, and EOEOEA 60% gel electrolytes at 30 °C was 5.60 × 10^−4^, 4.81 × 10^−4^, 4.00 × 10^−4^, and 3.42 × 10^−4^ S/cm, much higher than that of PEO-based solid polymer electrolytes (1.0 × 10^−8^ s/cm) and lower than that of a traditional liquid lithium battery (1.0 × 10^−3^ S/cm) by only one order of magnitude. These GPEs can meet the requirement for ionic conductivity of lithium batteries in practical applications [28]. 

The high conductivity of those EOEOEA GPEs should be attributed to the ester and ether groups in EOEOEA, resulting in a high solubility for lithium salts [29,30]. On the other hand, as shown in Figure 5a, at a certain temperature, the conductivity of gel electrolytes decreased with the increase of EOEOEA content, because a higher EOEOEA content led to a lower lithium salt content and lower conductivity [31,32]. For a certain GPE, the conductivity increases with the elevation of temperature, because a higher temperature prompts the movement of lithium salts; and, the changes in ionic conductivity against temperature agreed with the Arrhenius equation. It was calculated that the Ea of LiBF_4_-EOEOEA 30%, 40%, 50%, and 60% GPEs was 7.515, 9.994, 11.532, and 12.099 kJ/mol, respectively. The activation energy increased with the increase content of EOEOEA, because the hardness of gel electrolytes increased with the increase of EOEOEA content, which required more energy for the transmission of lithium ions in electrolytes.

Figure 5b shows the room-temperature electrochemical windows of GPEs with different EOEOEA contents. The windows of EOEOEA 30%, EOEOEA 40%, EOEOEA 50%, and EOEOEA 60% GPEs were wider than 4.65 V, meeting the requirement for electrochemical stability of high-voltage lithium batteries. All the widths were far greater than that (~3.9 V) of traditional liquid lithium batteries and PEO-based solid electrolytes [33]. Noticeably, the electrochemical window of LiBF_4_-EOEOEA 60% GPE was the widest (~5.05 V) among these GPEs, proving that the increase of EOEOEA content is favorable to the electrochemical stability of gel electrolytes. To sum up, EOEOEA is a gel electrolyte polymer substrate with great electrochemical stability.

### 3.4. XRD Results of LiBF_4_-EOEOEA-Based GPEs

Figure 6 shows the XRD patterns of LiBF_4_-EOEOEA 30% and 60% GPEs. LiBF_4_-EOEOEA 30% GPE only exhibited one diffraction spike at 2θ of 23°, indicating that the crystalline zone of LiBF_4_-EOEOEA 30% was very limited. With the increase of EOEOEA content, the crystalline zone of LiBF_4_-EOEOEA 60% GPE was extended, averse to the transmission of lithium ions and resulting in a decrease in conductivity. The XRD results are consistent with the results shown in Figure 5a, verifying that the ionic conductivity decreases when the EOEOEA content increases. 

### 3.5. Thermal Analysis of LiBF_4_-EOEOEA-Based GPEs

During charging and discharging cycles, lithium batteries should release a certain amount of heat, so GPE should possess good thermal stability. Figure 7a,b show TG and Derivative thermogravimetry ( DTG) curves for EOEOEA 30% GPE and EOEOEA 60% GPE, respectively. 

As illustrated in Figure 7a,b, LiBF_4_-EOEOEA 30% GPE lost weight at 312–506 °C, corresponding to the thermal decomposition of EOEOEA polymer. For LiBF_4_-EOEOEA 60% GPE, many weak signals were observed at 30–340 °C, which should be due to the decomposition of a larger amount of unreacted monomers in LiBF_4_-EOEOEA 60% GPE. Furthermore, a pronounced signal was observed at 340–530 °C for LiBF_4_-EOEOEA 60% GPE, similar to LiBF_4_-EOEOEA 30%. The decomposition temperature of LiBF_4_-EOEOEA 60% GPE was slightly higher than that of LiBF_4_-EOEOEA 30% GPE, indicating that the increase of EOEOEA content can increase the decomposition temperature of GPE.

### 3.6. Electrochemical Performances of LiBF_4_-EOEOEA-Based GPEs Lithium Batteries

Cycling performance is an important index in the quality of a battery. Figure 8a shows the specific discharge capacities of different EOEOEA-based GPE lithium batteries during 50 cycles under the condition of 0.1 °C, with a voltage range of 2.6–4.2 V. As illustrated in Figure 8a, the specific discharge capacities of LiBF_4_-EOEOEA 30%, 40%, 50%, and 60% lithium batteries in the first cycle were 116.45, 131.35, 110.47, and 86.00 mAh/g, respectively. The corresponding specific discharge capacities were still as high as 117.52, 129.36, 113.79, and 77.67 mAh/g, respectively, in the fiftieth cycle. The corresponding capacity retention rates after 50 cycles were 100%, 98%, 100%, and 90%, respectively. The 50 cycles of EOEOEA 30% and EOEOEA 40% lithium batteries seemed to be very stable, and the specific discharge capacities of both batteries were presented as straight horizontal lines. The specific discharge capacity of the EOEOEA 40% lithium battery was higher than that of the EOEOEA 30% lithium battery, showing that in a certain range, ionic conductivity cannot be the only factor affecting the specific discharge capacities of lithium batteries. Because the hardness of LiBF_4_-EOEOEA 40% GPE was slightly larger than that of LiBF_4_-EOEOEA 30% GPE, the former was able to realize a better contact of GPE with the positive and negative ions in the LiBF_4_-EOEOEA 40% GPE battery, so the ion migration ability at the interface was larger, which ultimately led to a better discharge performance compared to the LiBF_4_-EOEOEA 30% lithium battery [34]. The discharge specific capacity of EOEOEA 60% lithium battery is the smallest and the fluctuation is the largest, because the minimum conductivity of EOEOEA 60% gel electrolyte leads to incomplete charging and discharging of the battery, the discharge specific capacity is the smallest. At the same time, the hardness of LiBF_4_-EOEOEA 60% GPE is the largest; lithium ion transport activation energy becomes larger, and lithium ion transport becomes difficult, resulting in a relatively unstable interface chemical reaction, so LiBF_4_-EOEOEA 60% lithium battery discharge is unstable.

Figure 8b shows the Coulomb efficiencies of different EOEOEA-based GPEs lithium batteries during 50 charge-discharge cycles under the condition of 0.1 °C (the internal image is an enlarged version of the EOEOEA 30% GPE battery and the EOEOEA 40% GPE battery). The Coulomb efficiencies of LiBF_4_-EOEOEA 30%, 40%, 50%, and 60% GPE lithium batteries in the first cycle were 75%, 80%, 73%, and 81%, respectively. The Coulomb efficiencies of LiBF_4_-EOEOEA 30% and LiBF_4_-EOEOEA 40% batteries were presented as straight horizontal lines during the 50 cycles, indicating that the charge-discharge cycling processes of both batteries were very stable. In contrast, the Coulomb efficiencies of LiBF_4_-EOEOEA 50% and LiBF_4_-EOEOEA 60% lithium batteries fluctuated to a small extent, so not as stable as those of LiBF_4_-EOEOEA 30% and LiBF_4_-EOEOEA 40% batteries.

Figure 8c shows the specific charge and discharge capacities versus voltage relationships of the EOEOEA 30% lithium battery in the first, 25th, and 50th cycles. The charging platforms were located at about 3.4 and 3.6 V, which correspond to the insertion and removal processes of lithium ions in lithium iron phosphate electrodes [19]. The specific capacities of first discharge were 153.27 and 116.45 mAh/g, respectively. The Coulomb efficiency in the first cycle was 75%. In the 50th cycle, the specific discharge capacity was 117.52 mAh/g, comparable to the value in the first cycle, which proves that the charge-discharge cycling process of this battery was very stable and the capacity retention rate was as high as 100%. All the profiles in Figure 8c are very smooth without spikes, indicating that this type of GPE was very stable in the charge-discharge reactions and no side reactions occurred.

Figure 8d shows the specific discharge capacities of the LiBF_4_-EOEOEA 30% lithium battery at different rates. The values were 113.95, 105.39, 79.92, and 24.28 mAh/g under different C-rates of 0.1, 0.2, 0.5, and 1.0 °C, respectively. After charging and discharging cycles under the condition of high current density, the specific discharge capacity still reached 118.32 mAh/g under the condition of 0.1 °C, even slightly higher than that in the first discharging cycle, proving that this battery can bear high-current-density charging and discharging processes. 

Figure 8e shows the specific discharge capacities, and Figure 8f shows Coulomb efficiencies of the LiBF_4_-EOEOEA 30% GPE lithium battery and the liquid LiBF_4_ lithium battery. For comparison of charge-discharge stability, both batteries were subjected to 50 charge-discharge cycles under the condition of 0.1 °C. As shown in Figure 8e, the specific discharge capacity and stability of the LiBF_4_-EOEOEA 30% lithium battery were better than those of the traditional liquid lithium battery. In Figure 8f, the Coulomb efficiency of the LiBF_4_-EOEOEA 30% lithium battery was presented as a straight horizontal line, indicating the Coulomb efficiency was very stable. In contrast, the Coulomb efficiency of the traditional liquid LiBF_4_ battery fluctuated greatly, and the overall Coulomb efficiency was much lower. Thus, gel polymer lithium batteries will probably replace traditional liquid lithium batteries in the future.

Figure 9 shows the AC impedance plots of LiBF_4_-EOEOEA 30% and LiBF_4_-EOEOEA 60% lithium batteries before and after cycling. The AC impedance plots were treated with Zview software through fitting, and the bulk resistance values of LiBF_4_-EOEOEA 30% and LiBF_4_-EOEOEA 60% lithium batteries were 19.12 and 45.36 Ω, respectively, in great match with the ionic conductivity shown in Figure 5a. The diameters of the semicircles in the AC impedance plots represent the resistance values of lithium batteries. The resistance values of LiBF_4_-EOEOEA 30% and LiBF_4_-EOEOEA 60% lithium batteries before the cycling were 618.5 and 1137 Ω, respectively. Due to the passivation of lithium electrodes and gel electrolytes during the charging and discharging processes, the resistance values increased after 50 cycles. The values of LiBF_4_-EOEOEA 30% and LiBF_4_-EOEOEA 60% batteries were increased to 881.3 and 1663 Ω, respectively. The resistance of EOEOEA 30% was increased to a small extent, implying that the interface was relatively stable. In contrast, LiBF_4_-EOEOEA 60% contained a larger amount of unreacted monomers, and the internal passivation was more serious, increasing the contact resistance between gel polymer electrolyte and Li anode [6,35].

For learning the changes in internal components of the LiBF_4_-EOEOEA 30% battery after 50 cycles, the battery was disassembled in the glove box at 5.5 MPa with the sealer, and then the LiFePO_4_ electrode, cellulose composite membrane, and lithium electrode after the cycling were characterized by SEM. 

Figure 10a exhibits the original LiFePO_4_ electrode with an uneven surface and a maximum particle diameter of about 1 μm. Figure 10b shows the microscopic morphology of LiFePO_4_ electrode before the battery cycling. Clearly, the surface was coated with a layer of polymer electrolyte due to the occurrence of EOEOEA polymerization, and the surface was much smoother than the surface shown in Figure 10a. Figure 10c shows the LiFePO_4_ electrode after battery cycling. After 50 cycles, the GPE membrane still existed, and no small particles were found on the surface, indicating that side reactions did not occur on this GPE membrane, which possessed stable electrochemical performances. 

Figure 10d–f show the elemental mapping results of the surface of LiFePO_4_ electrode after 50 cycles. The surface of the LiFePO_4_ electrode contains the characteristic elements F (Figure 10e) and B (Figure 10f) of LiBF_4_-EOEOEA 30% GPE, which proves that the LiBF_4_-EOEOEA 30% precursor electrolyte can be polymerized on its surface, and the LiBF_4_-EOEOEA 30% GPE still remained after 50 cycles.

Figure 10g,h show the EDX results of the LiFePO_4_ electrode cross section after 50 cycles. At the interface between active species and Al foil, the characteristic elements F and B (Figure 10h) in LiBF_4_-EOEOEA 30% GPE were detected, showing that the LiBF_4_-EOEOEA 30% precursor electrolyte had completely penetrated into the LiFePO_4_ electrode and had polymerized to form gel electrolyte. It is conjectured that the gel electrolyte precursor had a good wetting performance on the LiFePO_4_ electrode. LiFePO_4_ can be completely wrapped by GPE by deep in-situ polymerization based on LiFePO_4_ electrode [18].

Figure 11 shows the XRD patterns of LiFePO_4_ electrode powder of the original, before and after 50 cycles. The XRD spikes of the original LiFePO_4_ electrode were observed at 2θ of 20.54°, 25.39°, 29.52°, 35.39°, and 36.36°, while these spikes were weakened or even disappeared for the spent LiFePO_4_ electrode. This is because the spent LiFePO_4_ electrode after cycling was tightly wrapped by the gel electrolyte, proving the occurrence of in-situ polymerization on the LiFePO_4_ electrode.

Because the solid content in a gel electrolyte is lower than that in a solid electrolyte, the mechanical strength of the former is not as high as that of the latter. Cellulose has a large length/diameter ratio, and in-situ polymerization on a cellulose membrane is beneficial to the mechanical strength of gel electrolyte composite membrane [36,37].

Figure 12a shows the original cellulose isolation membrane. The membrane was uneven on the surface, and contained many voids that were heterogeneously distributed. Figure 12b shows the composite isolation membrane of cellulose and gel electrolyte after the cycling. The voids have been completely filled, and the surface is smoother. After 50 cycles, no small particles were observed on the surface, which indicates that side reactions did not occur during the 50 cycles.

Figure 12c shows the SEM images of the original lithium electrode. The original lithium electrode is uneven on the surface. During the 50 cycles of traditional liquid lithium batteries, lithium dendrite grows. Some types of polymer electrolytes can inhibit the growth of lithium dendrite [38]. Figure 12d shows the SEM image of the lithium electrode after 50 cycles. The surface is smooth, and lithium nanoparticles are observed on the surface, indicating that this gel electrolyte can inhibit the growth of lithium dendrite [39].

## 4. Conclusions

By heating EOEOEA and LiBF_4_ electrolyte precursors via a thermal curing method, we prepared a family of gel polymer electrolytes with high ionic conductivity (5.6 × 10^−4^ S/cm) at 30 °C, and wide electrochemical windows (4.65 V) at room-temperature and good thermal stability. The ionic conductivity decreased as the EOEOEA content increased, whereas the electrochemical window and thermal stability increased as the EOEOEA content increased.

On the basis of the simple in-situ deep polymerization on the LiFePO_4_ electrode and cellulose membrane in the battery container on the cathode side, GPE generated on both the LiFePO_4_ electrode and cellulose membrane. Meanwhile, the contact between the gel electrolyte with LiFePO_4_ electrode and Li electrode, respectively, was enhanced, and the stability of gel electrolyte lithium was also enhanced. In addition, the growth of lithium dendrite was effectively inhibited. The capacity retention rate of the LiBF_4_-EOEOEA 30% lithium battery was 100% after 50 cycles under the condition of 0.1 °C, and the Coulomb efficiency surpassed 99%. This preparation approach provides scaffolds for the application of high-performance gel polymer electrolyte lithium batteries.

## Figures and Tables

**Figure 1 polymers-11-01296-f001:**
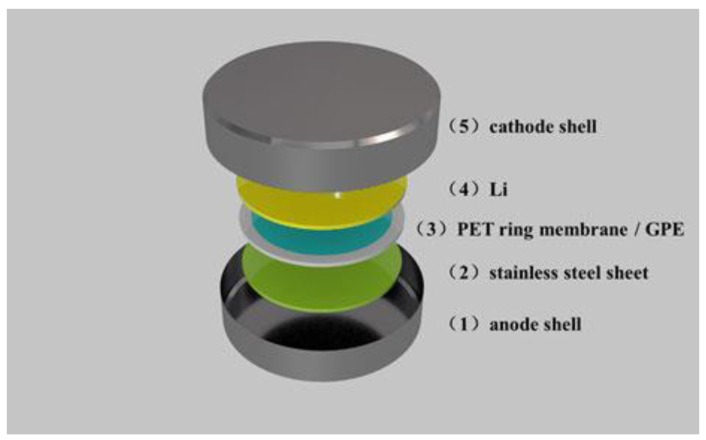
Structure of the stainless steel sheet/gel polymer electrolyte (GPE)/Li button battery.

**Figure 2 polymers-11-01296-f002:**
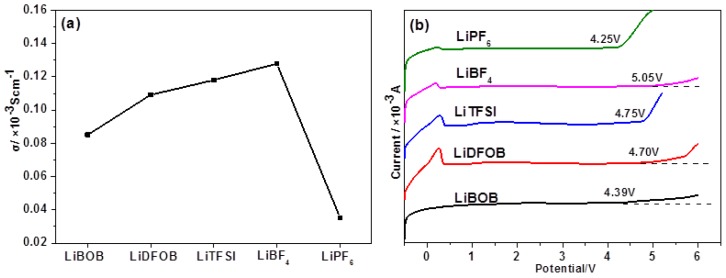
(**a**) Ionic conductivity at 30 °C and (**b**) room-temperature electrochemical windows of different lithium salt-EOEOEA 50% GPEs.

**Figure 3 polymers-11-01296-f003:**
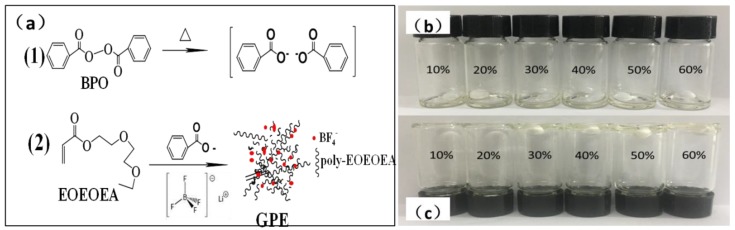
(**a**) The polymerization of EOEOEA initiated by BPO; precursor electrolytes with different contents of EOEOEA before (**b**) and after (**c**) heating.

**Figure 4 polymers-11-01296-f004:**
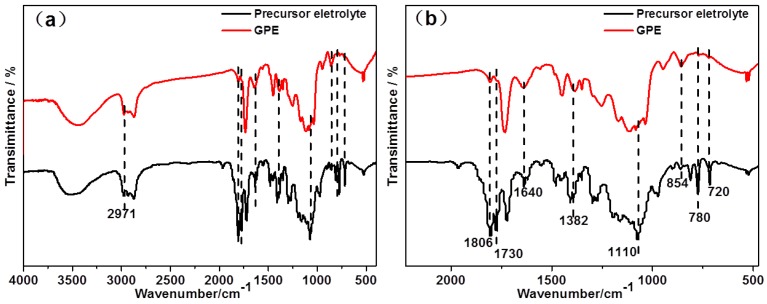
FTIR spectra of LiBF_4_-EOEOEA 30% precursor electrolyte and LiBF_4_-EOEOEA 30% GPE.

**Figure 5 polymers-11-01296-f005:**
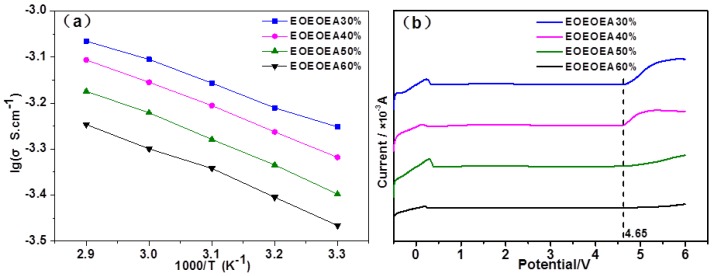
(**a**) Ionic conductivity at 30 °C and (**b**) room-temperature electrochemical windows of different LiBF_4_-EOEOEA GPEs.

**Figure 6 polymers-11-01296-f006:**
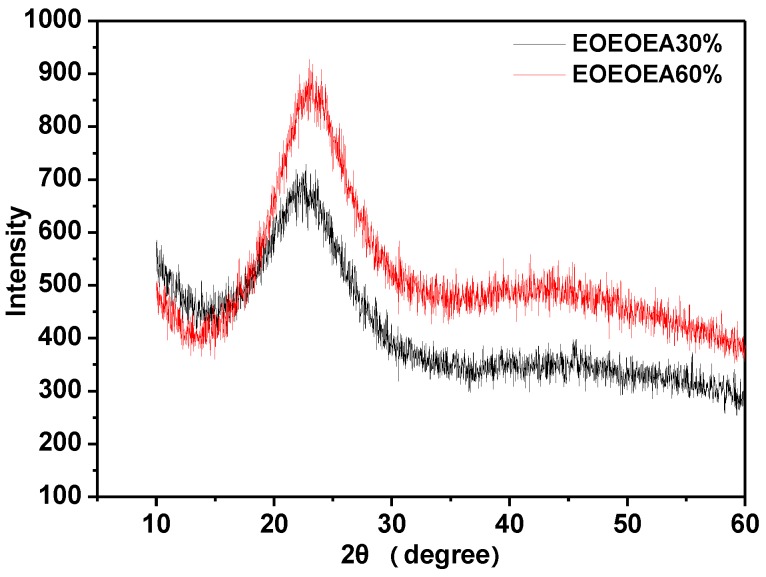
XRD patterns of different gel electrolytes.

**Figure 7 polymers-11-01296-f007:**
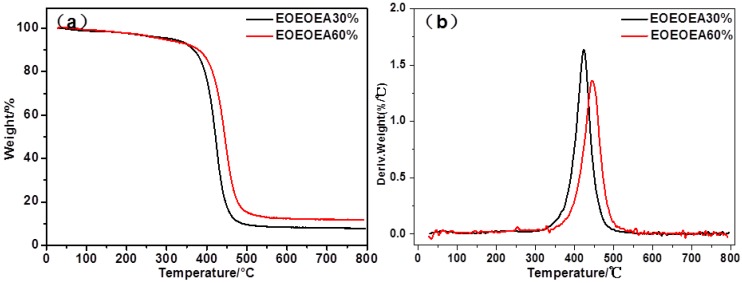
(**a**) TG and (**b**) DTG curves of different gel electrolytes.

**Figure 8 polymers-11-01296-f008:**
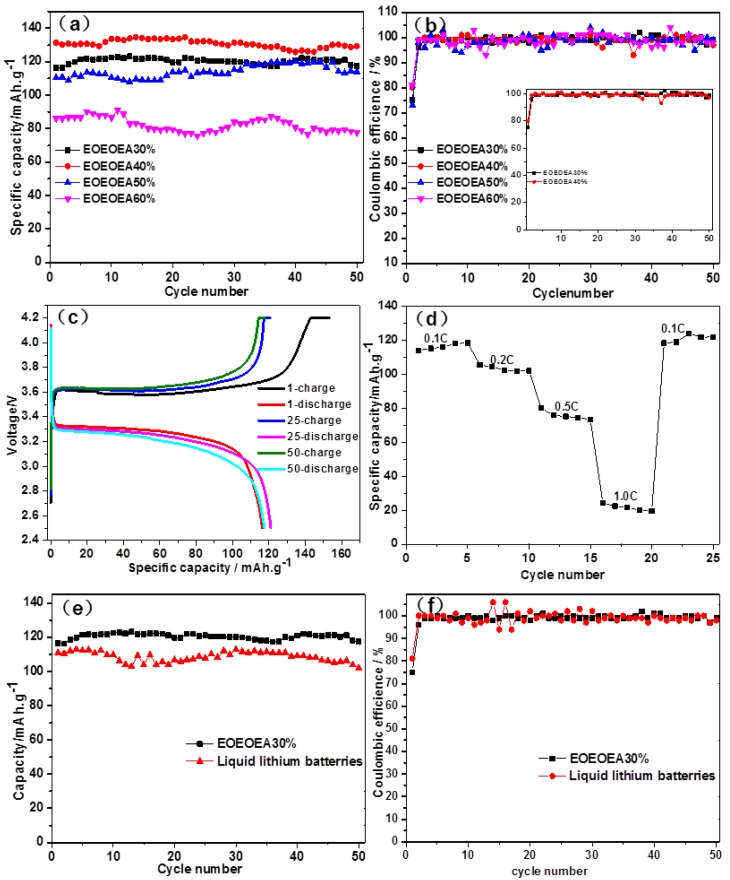
(**a**) Specific discharge capacities and (**b**) Coulomb efficiencies of different EOEOEA lithium batteries; (**c**) specific discharge capacities of the LiBF_4_-EOEOEA 30% lithium battery in the first, 25th, and 50th cycles; (**d**) discharging cycles of the LiBF_4_-EOEOEA 30% lithium battery at different rates; (**e**) specific discharge capacities and (**f**) Coulomb efficiencies of the LiBF_4_-EOEOEA 30% lithium battery and the liquid LiBF_4_ lithium battery.

**Figure 9 polymers-11-01296-f009:**
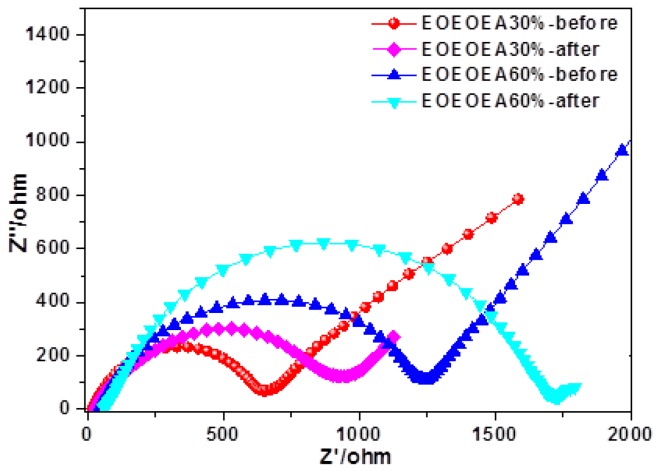
AC impedance plots of LiBF_4_-EOEOEA 30% and LiBF_4_-EOEOEA 60% gel lithium batteries before and after 50 cycles.

**Figure 10 polymers-11-01296-f010:**
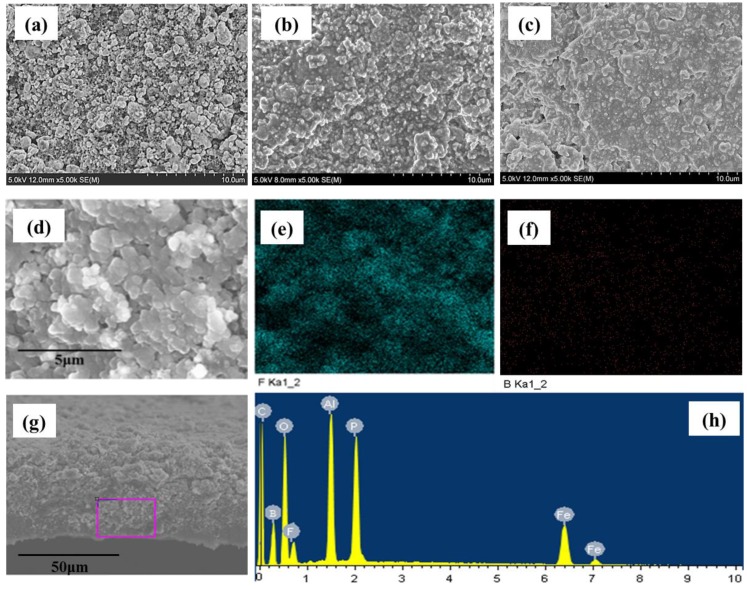
SEM images of the surfaces of (**a**) the original LiFePO_4_ electrode, (**b**) the LiFePO_4_ electrode before the cycling, and (**c**) the LiFePO_4_ electrode after 50 cycles. Elemental mapping results of the surface (**d**–**f**) and EDX results of the cross section (**g**,**h**) of LiFePO_4_ electrode after 50 cycles.

**Figure 11 polymers-11-01296-f011:**
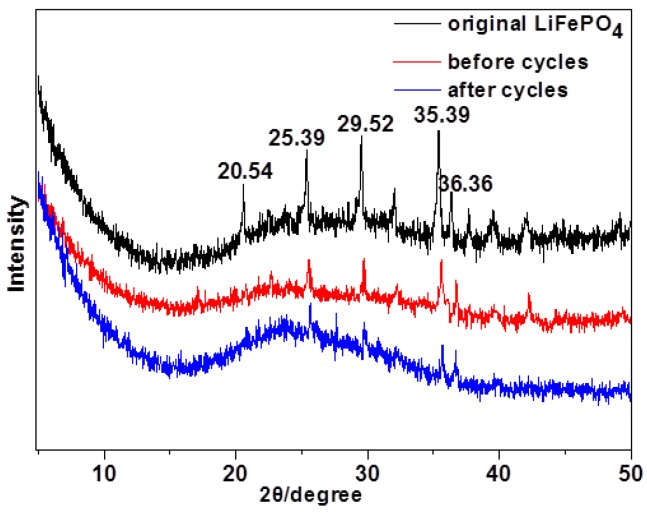
XRD patterns of LiFePO_4_ electrode powder of original, before and after 50 cycles.

**Figure 12 polymers-11-01296-f012:**
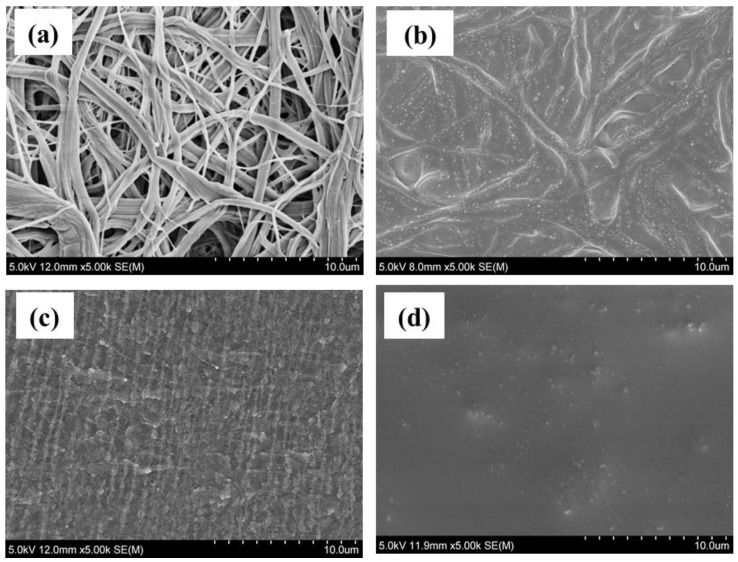
SEM images of (**a**) the original cellulose membrane and (**b**) the cellulose composite isolation membrane after 50 cycles, and (**c**) the original lithium electrode and (**d**) the lithium electrode after 50 cycles.

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
