# Peer review of "Preparation and Properties of a High-Performance EOEOEA-Based Gel-Polymer-Electrolyte Lithium Battery"

_polymers, 2019, doi:10.3390/polym11081296_

Round 1

Reviewer 1 Report

·        The quality of Figure 1 is very poor, try to provide better resolution figure.

·        In the conductivity plot LiBF4 salt provided better conductivity than LiTFSI, compared to BF4 anion. But as per my opinion, TFSI can make weak cation-anion interaction and improve the diffusion properties of the electrolyte than that of BF4- anion. The author should provide proper explanation/mechanism for LiBF4-polymer complexes and also why it reduced for LiTFSI-polymer complex?

·        The author should provide the loading mass of the active material?

·        The Y-axis scale is missing in Figure 2(LSV)? Why there is a small hump around 0.5V? Anodic oxidation is predominant? If so why not for LiBOB?

·         What are electrodes used for the linear sweep voltammetry?

·        The results and discussion for FT-IR should be modified, couldn’t make any impact on the formation of polymer complexes.

·        In the FT-IR spectra, the Y-axis scale should be transmittance. Also, the authors enlarge spectra in the region 2000-800 cm-1 in order to identify the characteristic peaks clearly.

·        The author has to cite recent bibliographies on gel polymer electrolytes in the introduction section to strengthen the obtained observations

10.1002/anie.201504971;10.1038/s41598-017-11614-1; 10.1016/j.memsci.2015.11.007; 10.1039/C7RA01081H; 10.1039/C7EE01095H; 10.1007/s10008-016-3466-2;  

·        In the TGA discussion part, the authors have mentioned that weight loss has occurred at 312 ºC about 30%. But they need to provide the temperature range instead of specific temperature

Reviewer 2 Report

The manuscript “Preparation and properties of a high-performance EOEOEA-based gel-polymer-electrolyte lithium battery” reported gel polymer electrolytes based on heating 2-(2-ethoxyethoxy)ethyl acrylate (EOEOEA) and LiBF4 electrolyte precursors via a thermal curing method, which showed high ionic conductivity at 30 ℃ , wide electrochemical window at room-temperature and good thermal stability . The authors have provided solid data to back up the conclusions in most cases. However, when reading the manuscript some questions arise, therefore some complementary information and revision should be taken into account before being published in “Polymers”.

1.     A schematic diagram for the chemical structure of all the major compounds used in the manuscript is suggested, such as EOEOEA, BPO, and all the lithium salts being used.

2.     In section 3.1, which EOEOEA content was used? 30%? In the following sections, which lithium salt was used? LiBF4? Such key Information should be defined clearly before discussion.

3.     In line 297-300, the authors claimed that “The specific discharge capacity of EOEOEA40% lithium battery was higher than that of EOEOEA30% lithium battery, showing that in a certain range, ionic conductivity could not be the only factor affecting the specific discharge capacities of lithium batteries.” The other possible factors leading to this result should be further discussed.

4.     Please add vertical scale for Figure 8b like Figure 8f or use different layers.

5.     Two different abbreviations for figures were used in the manuscript, “Fig.” and “Figure”. Please unify them.

6.     How the liquid LiBF4 lithium battery was assembled should be provided.

7.     In Figure 10g, please highlight the peaks of B and F since they are not clearly shown.

8.     In Figure 10 and 11, the characterization of the LiFePO4 electrode dissembled from fresh EOEOEA30% battery is also necessary to show the difference between original LiFePO4 electrode and EOEOEA30% LiFePO4 electrode after initial polymerization, as well as the EOEOEA30% LiFePO4 electrode before and after cycling.

9.     There are some other recently outstanding research works about all-solid-state electrolytes that the authors did not referenced, such as Tong, Yongfen, et al. "All-solid-state interpenetrating network polymer electrolytes for long cycle life of lithium metal batteries." Journal of Materials Chemistry A 6.30 (2018): 14847-14855.; Zeng, Xian-Xiang, et al. "Reshaping lithium plating/stripping behavior via bifunctional polymer electrolyte for room-temperature solid Li metal batteries." Journal of the American Chemical Society 138.49 (2016): 15825-15828; and so on, which are also worthy to reference.

10.  The English writing in manuscript needs to be checked very carefully before submission since the meaning of many sentences cannot be understood by the improper use of English or ambiguous description.

Round 2

Reviewer 2 Report

All the questions and comments raised by the reviewers were legitimately explained and revised. The accuracy and detail of the manuscript were also improved further after revision. Therefore, the revised manuscript is now suitable publication in "Polymers".